# PD-L1 Expression in High-Risk Early-Stage Colorectal Cancer—Its Clinical and Biological Significance in Immune Microenvironment

**DOI:** 10.3390/ijms232113277

**Published:** 2022-10-31

**Authors:** Bing-Syuan Chung, I-Chuang Liao, Peng-Chan Lin, Shang-Yin Wu, Jui-Wen Kang, Bo-Wen Lin, Po-Chuan Chen, Ren-Hao Chan, Chung-Ta Lee, Meng-Ru Shen, Shang-Hung Chen, Yu-Min Yeh

**Affiliations:** 1Department of Internal Medicine, National Cheng Kung University Hospital, College of Medicine, National Cheng Kung University, Tainan 70456, Taiwan; 2Department of Pathology, Chi Mei Medical Center, Tainan 70456, Taiwan; 3Department of Oncology, National Cheng Kung University Hospital, College of Medicine, National Cheng Kung University, Tainan 70456, Taiwan; 4Department of Genomic Medicine, National Cheng Kung University Hospital, College of Medicine, National Cheng Kung University, Tainan 70456, Taiwan; 5Department of Computer Science and Information Engineering, College of Electrical Engineering and Computer Science, National Cheng Kung University, Tainan 70456, Taiwan; 6Division of Colorectal Surgery, Department of Surgery, National Cheng Kung University Hospital, College of Medicine, National Cheng Kung University, Tainan 70456, Taiwan; 7Department of Pathology, National Cheng Kung University Hospital, College of Medicine, National Cheng Kung University, Tainan 70456, Taiwan; 8Department of Obstetrics and Gynecology, National Cheng Kung University Hospital, College of Medicine, National Cheng Kung University, Tainan 70456, Taiwan; 9National Institute of Cancer Research, National Health Research Institutes, Tainan 70456, Taiwan

**Keywords:** PD-L1, combined positive score, colorectal cancer, prognostic biomarker, CXCL9

## Abstract

Programmed death-ligand 1 (PD-L1) is an immune checkpoint molecule that can regulate immune responses in the tumor microenvironment (TME); however, the clinical applications of PD-L1 in early-stage colorectal cancer (CRC) remain unclear. In this study, we aimed to investigate the relationship between PD-L1 expression and survival outcome and explore its relevant immune responses in CRC. PD-L1 expression was evaluated by immunohistochemical staining to determine the tumor proportion score and combined positive score (CPS) in a Taiwanese CRC cohort. The oncomine immune response research assay was conducted for immune gene expression analyses. CRC datasets from the TCGA database were reappraised for PD-L1-associated gene enrichment analyses using GSEA. The high expression of PD-L1 (CPS ≥ 5) was associated with longer recurrence-free survival (*p* = 0.031) and was an independent prognostic factor as revealed by multivariate analysis. High PD-L1 expression was related to six immune-related gene signatures, and *CXCL9* is the most significant overexpressed gene in differential analyses. High *CXCL9* expression correlated with increased infiltration levels of immune cells in the TME, including CD8+ T lymphocytes and M1 macrophages. These findings suggest that high PD-L1 expression is a prognostic factor of early-stage CRC, and CXCL9 may play a key role in regulating PD-L1 expression.

## 1. Introduction

Colorectal cancer (CRC) is one of the most prevalent cancers around the world. In Taiwan, nearly 16,000 cases of CRC were newly diagnosed in 2019, and about 80% of them were regional or locally advanced CRCs [1]. Approximately 6000 patients died from this disease in the same year in Taiwan. Regular screening plays an important role in CRC management because the survival outcome differs greatly between early and late-stage CRCs. Even though early-stage CRCs have a relatively good prognosis with a 5-year survival of 72–91% [2], high-risk stage II and stage III CRCs are the subgroups with the poorest survival. Only about 59% of patients in these subgroups would achieve disease-free status by undergoing curative surgery alone [3]; so, curative surgery following adjuvant chemotherapy is the standard treatment. Although adjuvant chemotherapy improves the 3-year disease-free survival to 78.2% in these subgroups [4], a certain proportion of patients cannot receive any benefit from adjuvant chemotherapy. How to precisely detect which patient needs adjuvant chemotherapy is still an unresolved issue. Many classification methods and prognostic biomarkers have been proposed for evaluating survival in early-stage CRCs; however, they are yet to become the guidance in clinical practice because of many factors, such as feasibility, reproducibility, and accuracy.

PD-L1, also known as *CD274*, is a transmembrane checkpoint protein expressed on various types of immune and tumor cells. The PD-L1/PD-1 pathway downregulates T-cell function during inflammatory states and takes part in adaptive immune resistance in cancer [5]. PD-L1 expression has a prognostic value in various types of cancer [6,7,8,9,10,11,12,13,14]; however, its significance in early-stage CRCs is still debated. The inconsistent results are probably from the heterogeneity of the study population, different PD-L1 staining antibodies, different definitions of PD-L1 positivity, etc. However, PD-L1 immunohistochemistry has standardized guidance for staining protocol and interpretation criteria. As a prognostic marker, PD-L1 staining is also convenient and relatively cost effective. Owing to more efforts exploring the utility of PD-L1 as a prognostic marker, studies demonstrated that PD-L1 expressed on tumor cells or tumor-infiltrating cells might result from separate mechanistic pathways and affect survival outcomes differently. [13,14,15]. The regulations of PD-L1 expression are complicated, including genomic, epigenetic, transcriptional, and posttranscriptional levels; however, its detailed mechanisms are yet to be elucidated in the CRC microenvironment [16]. A better understanding of mechanisms regulating PD-L1 expression in the tumor microenvironment (TME) may help us clarify the clinical utility of PD-L1 expression or even immunotherapy-based treatments in CRC. Therefore, this study aimed to evaluate the prognostic value of PD-L1 expression in high-risk, early-stage CRCs and to explore the possible underlying mechanisms.

## 2. Results

### 2.1. Clinical Characteristics of the Study Participants

A total of 100 patients, including 3 patients with high-risk stage II and 97 patients with stage III CRC, were enrolled. Until the end of 2021, the median follow-up of this CRC cohort was 5.2 years. Recurrence was detected in 32 of 100 patients and 68 patients remained in disease-free states. The clinical characteristics of these 100 patients are shown in Table 1. The median age of these patients was 56.5 years, 54% were male, and 23% had right-sided colon cancer. The majority of tumors were adenocarcinomas (91%) and moderately differentiated (91%). The *RAS* mutation, *BRAF* mutation, and deficient mismatch repair/microsatellite instability-high (dMMR/MSI-H) were detected in 42%, 6%, and 6% of tumors, respectively. PD-L1 expression in primary CRC tumor samples was determined by the tumor proportion score (TPS) and combined positive score (CPS). Figure 1 shows the representative immunostaining results of CRC with a CPS score of <1 (Figure 1A), 1–4 (Figure 1B), 5–9 (Figure 1C), and ≥10 (Figure 1D). When PD-L1 expression was evaluated by TPS, most of the tumors had no or very low PD-L1 expression (<1%), and only 5% of tumors had TPS of ≥1%. When CPS was used to assess the PD-L1 expression, 53% of tumors had positive PD-L1 expression, which was defined by CPS of ≥1, and 47% of tumors had no PD-L1 expression (CPS < 1). CPS 1–4, 5–9, and ≥10 accounted for 26%, 15%, and 12% of the entire cohort. By using the double staining technique, the main proportion of PD-L1 expression was identified on CD68+ macrophages and a small proportion was on CD3+ T lymphocytes in the tumor tissue with high CPS (>10) as shown in Appendix A.

### 2.2. Survival Analysis by Different Levels of PD-L1 Expression

To investigate the prognostic value of PD-L1 expression, we performed a cut point analysis for RFS, overall survival (OS), and hazard ratios across different scoring methods and different levels of PD-L1 expression. When TPS was applied to determine the PD-L1 expression, there was no survival difference between patients with TPS of ≥1% and TPS of <1%, as shown in Figure 2A. It was noteworthy that when CPS of ≥3 or 5 was used to categorize the patients as high PD-L1 expression, patients with high PD-L1 expression had significantly better RFS than those with low PD-L1 expression (*p* = 0.038 and *p* = 0.031, respectively; Figure 2B,C). There was also a trend showing a better OS for the high PD-L1 expression group compared with those with low PD-L1 expression, especially for patients with CPS of ≥5. These results indicate that CPS can be a prognostic marker for patients with early-stage CRC.

### 2.3. Clinicopathological Features of Patients with High and Low PD-L1 Expression

When CPS of ≥5 was used to define the high expression of PD-L1, patients with high PD-L1 expression were significantly more likely to have right-sided (*p* = 0.046) and dMMR/MSI-H CRC (*p* = 0.024) compared to patients with low PD-L1 expression (Table 2). There was no significant difference between patients with high and low PD-L1 expression in terms of age, gender, tumor histology, tumor grade, RAS mutation status, and *BRAF* mutation status. The univariate analysis showed that mutant RAS and low PD-L1 expression were associated with worse RFS (Table 3). In the multivariate analysis, PD-L1 expression was still an independent prognostic factor for RFS.

### 2.4. Immune Pathways Associated with High PD-L1 Expression

To explore the potential pathways associated with PD-L1 expression, mRNA expression data were downloaded from the TCGA PanCancer Atlas database. After data cleaning, a total of 184 CRCs, including 99 stage II and 85 stage III CRCs, were enrolled for analysis. Detailed clinical characteristics provided by the TCGA database are listed in Appendix A. When hallmark gene sets were applied for the GSEA analysis, 15 gene signatures were statistically enriched in the high *CD274* expression group (Appendix A). Among these gene signatures, interferon (IFN)-gamma response, IL-2-STAT5 signaling, IL-6-STAT3 signaling, inflammatory response, TNFA signaling via NFKB, and IFN-alpha response are immune-related pathways (Figure 3). Moreover, *CD274* expression was positively correlated with *CD3G* (r = 0.720; *p* < 0.001) and *CD68* (r = 0.555; *p* < 0.001) expression, respectively, using the TCGA database (Appendix A). These findings suggest that PD-L1-associated immune responses may be involved in the clinical outcomes of patients with early CRC.

### 2.5. Immune-Related Genes Differentially Expressed between PD-L1 High and Low CRCs

To further understand the tumor immune microenvironment contributing to the difference in outcome between CRC patients with high and low PD-L1 expression, the oncomine immune response research assay, a targeted NGS assay analyzing the expression of 395 genes associated with the immune response, was applied on the primary CRC tumor samples. The differential gene expressions were compared between patients with high (CPS ≥ 5) and low PD-L1 expression (CPS < 5). In this CRC cohort, the gene expression levels were available in 70 patients, including 18 patients with high PD-L1 expression and 52 patients with low PD-L1 expression. As shown in Figure 4A,B, a total of 12 genes were found to be differentially expressed between the two groups (adjusted *p* < 0.1). The expression levels of *CXCL9*, *PRF1*, *PDCD1*, *SIT1*, *BST2*, and *ISG15* were higher in CRCs with high PD-L1 expression compared to those with low PD-L1 expression. In contrast, CRCs with low PD-L1 expression had higher expression levels of *CDKN3*, *CD44*, *ABCF1*, *MAPK1*, *RPS6*, and *LAMP1* than those with high PD-L1 expression. The prognostic value of these genes reported in the literature and relevant references are listed in Figure 4B. Among these genes, *CXCL9* was the top gene differentially expressed between these two groups of CRCs and can encode the chemokine, CXCL9, known to be regulated by IFN-gamma [17]. When the gene expression level of PD-L1 was analyzed, we also observed a positive correlation between *CXCL9* expression and *CD274* expression (Figure 4C) in our CRC cohort, which was supported by the data of colon cancer (Figure 4D) and rectal cancer cohort from TCGA (Figure 4E).

### 2.6. Correlation between CXCL9 Expression and Immune Cell Infiltrations

CXCL9 plays an important role in regulating immune cell migration, differentiation, and activation of the TME [17]. Since *CXCL9* was differentially expressed between CRCs with high and low PD-L1 expression, we further analyzed the impact of *CXCL9* on immune cell infiltrations. CRCs from the TCGA database were categorized into the group of high and low *CXCL9* expression according to the median expression value of *CXCL9*, and immune cell infiltrations were compared between these two groups by using the TIMER 2.0. As shown in Figure 5, B-cell infiltration was lower in CRCs with high *CXCL9* expression compared to those with low *CXCL9* expression. CRCs with high *CXCL9* expression had more CD8+ T cells and less regulatory T-cell infiltration than those with low *CXCL9* expression. The infiltrations of M1 macrophages, M2 macrophages, and cancer-associated fibroblasts were also higher in CRCs with high *CXCL9* expression than in those with low *CXCL9* expression. No significant difference in CD4+ T-cell and neutrophil infiltration was found between these two groups.

## 3. Discussion

In this study, we analyzed the prognostic impact of PD-L1 expression in patients with stage III and high-risk stage II CRC, and the potential interactions in the tumor immune microenvironment. Our data showed that CPS of ≥5 was associated with better RFS in early-stage CRC patients undergoing the standard surgery and receiving the adjuvant FOLFOX chemotherapy. CRCs with high PD-L1 expression had a higher expression level of *CXCL9*. The data from the TCGA PanCancer Atlas showed that IFN-gamma signaling was one of the immune-related gene signatures enriched in CRCs with high PD-L1 expression, and high *CXCL9* expression was associated with more CD8+ T cells and M1 macrophages but less regulatory T-cell infiltration.

The role of PD-L1 expression as a prognostic marker for survival in early-stage CRC is controversial. This controversy may result from several factors, including the heterogeneity of the study populations, various IHC staining methods, different cutoff values for defining PD-L1 positivity, and treatment modalities. Although the prognostic value of PD-L1 expression remains inconclusive, several studies have reported that PD-L1 expression is an important prognostic factor of CRC. Recent studies have demonstrated that higher PD-L1 expression on immune cells in the TME is correlated with better survival outcomes in early-stage CRC [6,8,9,33]. On the contrary, some clinical studies have reported that prominent PD-L1 expression on tumor cells indicates poorer survival outcomes of CRC. [7,8,11,12]. Notably, a few reports have shown that PD-L1 expression is a prognostic factor for CRC in a site-dependent manner [8,13]. PD-L1 expression on tumor or immune cells can have distinct survival outcomes for patients with CRC. Because of these divergent findings, the clinical utility of PD-L1 expression as a prognostic marker remains uncertain in CRC.

In the past decades, scientists have been eager to find practical prognostic factors for early-stage CRC patients. Several efforts have been made to determine which patients should receive adjuvant chemotherapy or which should be closely monitored. Although many classification methods have been introduced, such as CMS, MSI, and the immune score, none of them have been applied in real-world clinical practice. The circulating tumor DNA (ctDNA)-guided approach is a promising method that has been proven to prevent unnecessary adjuvant chemotherapy [34]. However, the financial burden of applying ctDNA to daily practice would still be a potential hurdle to overcome in the future. On the contrary, our findings showed that CPS, a long-established biomarker with standardized methodology, has good prognostic value in early-stage CRCs. In the present study, early-stage CRCs with high expression of PD-L1 (especially CPS ≥ 5) were significantly associated with better RFS. CPS itself was also an independent prognostic factor of RFS using univariate and multivariate analyses. In current clinical practice, the CPS score, which is defined as the number of PD-L1-positive cells, including tumor and immune cells, divided by the total number of tumor cells × 100, has been an approved biomarker for predicting the efficacy of anti-PD-1/PDL-1 antibodies (the so-called immune-checkpoint inhibitor (ICI)) in various types of cancer [35,36]. This method of evaluating PD-L1 expression in the TME is a standardized and easily-accessible approach for clinical physicians. Collectively, our findings suggest the clinical utility of PD-L1 expression as a prognostic marker for patients with early CRC.

For further investigation of possible associated mechanisms of the regulation of PD-L1 expression, we used the GSEA to analyze different enriched pathways between early-stage CRCs with high and low *CD274* expression from the TCGA database. Fifteen gene signatures were enriched in the higher *CD274* expression group, and several enriched pathways are cancer-associated immune responses. The role of these immune-related pathways in the carcinogenesis in CRC have been extensively studied. Generally, IL-2 [37,38], IFN-gamma [39], IFN-alpha [40], and TNF-alpha signaling [41] can elicit a pro-inflammatory response and demonstrate potent anti-tumor effects in CRC through a direct and indirect mechanism. In parallel with the anti-tumor immunities, clinical studies have showed the positive correlation between enrichment of these immune-related pathways and better clinical outcomes of CRC [42,43,44]. These results can support our findings that high-CPS is a prognostic factor for longer survivals of patients with CRC. In contrast, IL-6 is a cytokine well known to be involved in the development and progression of many cancer types, including CRC [45]. In addition to the direct promoting effects on tumor cells, IL-6 could induce CRC progression by modulating the tumor immune microenvironment [37,38]. Prior studies have showed that high IL-6 levels are associated with an advanced stage, high risk of relapse, and worse survival outcomes of CRC [46,47,48]. Several studies have explored the PD-L1 expression mediated by IL-6 in certain cancer types [49,50]. Understanding the interaction between PD-L1 and IL-6 in the tumor immune microenvironment would provide novel insights of therapeutic strategies against CRC.

These results were consistent with findings from immune gene profiling from our National Cheng Kung University Hospital (NCKUH) cohort, in which CXCL9, a chemokine regulating immune cell migration, differentiation, multiplication, and activation, was also upregulated in the high CPS expression group. According to our GSEA analysis results, CXCL9 took part in three of the immune response-enriched gene signatures, including IFN-gamma, IL-6, and inflammatory response pathways. CXCL9 secretion can be induced by IFN-gamma stimulation, and this IFN-gamma-CXCL9 pathway has an essential role in regulating tumor growth [17,51]. In general, IFN-gamma-JAK-STAT1 signaling is the most well-known pathway that regulates PD-L1 expression in the TME. However, several studies have shown that PD-L1 expression can be upregulated on various types of tumor cells by the activation of the STAT3 and PI3K-Akt pathways when CXCL9 is stimulated by IFN-gamma signaling [52,53]. While CXCL9 upregulates and recruits immune cells in the TME to inhibit tumor growth, it also induces PD-L1 expression to help tumors escape immune cell surveillance. Tokunaga et al. proposed the idea that CXCL9 has two types of signaling methods in regulating the TME. One is through paracrine signaling from monocytes, endothelial cells, and fibroblasts, and the other is through autocrine signaling from cancer cells [17]. These two signaling interactions have totally different impacts on tumor growth: CXCL9 with paracrine signaling mainly has tumor-suppression effects, and CXCL9 with autocrine signaling causes cancer cell proliferation and metastasis. These complicated and contradictory roles of CXCL9 on tumor growth may be related to the intricate spatial and temporal interaction of immune responses in the TME, not simply the location where CXCL9 is secreted.

According to our results, *CD274* expression is positively correlated with *CXCL9* expression. Accumulating studies have demonstrated that high expression of *CXCL9* is associated with better survival outcomes of CRC [54,55]. These results can support the rationale of PD-L1 expression as a potential prognostic factor of CRC. Additionally, CXCL9 is a crucial cytokine regulating immune cell migration, differentiation, multiplication, and activation, and the lack of CXCL9 can cause the failure of effector T-cell trafficking to TME [56,57]. CD103+ dendritic cells (DCs) are the primary source of CXCL9 secretion in the TME. After stimulation by IFN-gamma, DC-derived CXCL9 increases and reactivates CD8+ T cells. Subsequently, more IFN-gamma can be produced by CD8+ T cells and a positive feedback loop intensifying antitumor immune response can be formed [58]. In addition to T-cell migration and activation, DCsre associated with the efficacy of anti-PD1 therapies through intratumoral CD8+ T-cell proliferation [18]. Therefore, the TME of early-stage CRC is notably different between the high and low *CXCL9* expression groups. The high *CXCL9* group has increased infiltration levels of CD8+ T cells and macrophages. Because CD8+ T-cell infiltration in TME is a well-established prognostic factor across various types of cancer, it may partly explain why patients with early-stage CRC showing high CPS expression have better survival benefits compared to their low CPS expression counterparts [59,60,61,62,63]. In addition to prognostic significance, this evidence may suggest a potential therapeutic approach to CRC through the combination of ICIs and CXCL9-based therapy. The efficacy of anti-PD-1/PD-L1 antibodies has been well demonstrated in various types of cancer, including MSI-H CRC [64,65]. However, dMMR/MSI-H CRC comprises approximately 5% of metastatic CRC, and the efficacy of ICIs has been unsatisfactory in the majority of CRC which is mismatch repair proficient (pMMR) or microsatellite stable (MSS) [66,67]. Because MSS metastatic CRC is usually classified as a typical “cold” cancer that presents a lack of the activation of immune responses, the studies investigating how to generate a “hot” TME are critical to the treatment improvement of MSS CRC [68,69]. The production of CXCL9 in the TME can induce T-cell infiltration and may contribute to the orchestration of the “hot” TME of MSS CRC. Several preclinical studies have shown that the efficacy of immunotherapeutic strategies can be enhanced through the manipulation of CXCL9-involving immune interactions [70,71]. In clinical studies, therapeutic interventions targeting toll-like receptors (TLRs), which are involved in the innate immune system, can induce CXCL9 production in the immune microenvironment [55,72]. Some early-phase clinical trials using a combination strategy with TLR agonists and pembrolizumab (anti-PD-1 antibody) in various types of cancer are ongoing, including pMMR CRC (NCT02834052). The results of these clinical trials are notable because of their potential for advancing clinical applications of immunotherapy in CRC.

This present study combined a Taiwanese cohort and the TCGA database for the analysis. The population of the NCKUH cohort is homogenous, which is purely high-risk stage II and stage III patients who underwent curative surgery and received adjuvant FOLFOX chemotherapy in a single medical center. The survival analysis would be more reliable than pure online database research. However, our study had some limitations. First, this is a retrospective study, which means there might have been selection bias. Second, we only used one IHC staining method (Dako 22C3) to evaluate PD-L1 expression. Potential discordance between different antibodies may affect subsequent analytic findings. Moreover, different cutoff values are applied to determine PD-L1 positivity in different studies, which would also lead to inconsistent findings. Third, the analysis of possible pathways and the TME behind high PD-L1 or CXCL9 expression is mainly based on in silico studies. These studies identified a few immunosuppressive cell types correlated with CXCL9 expression in the TME, such as M2 macrophages and cancer-associated fibroblasts. Although accumulating evidence has demonstrated that CXCL9 expression in response to IFN-gamma signaling can exert antitumor activity through increased infiltration of effector T lymphocytes, the subsequent CXCR3 activation may be involved in macrophage or stroma cell polarization of the TME [73,74]. The biological processes in regulation of tissue inflammation and wound healing may contribute to versatile functions of these immune molecules. Cancer cells may take advantage of biological processes maintaining physical function to facilitate tumor growth. For advanced understanding the CXCL9-PD-L1 immune interactions in TME of CRC, additional biological experiments and mechanistic studies are warranted.

In summary, our data revealed that PD-L1 expression is a practical prognostic marker for RFS in patients with high-risk early-stage CRC. In the TME of CRC, high PD-L1 expression is associated with crucial cancer-associated immune responses, including IFN-gamma response, IL-2-STAT5 signaling, and the IL-6-STAT3 signaling pathway. Moreover, CXCL9 is an important mediator involved in immune interactions upregulating PD-L1 expression and the activation of various types of immune cells, such as CD8+ T cells and M1 macrophages. These findings may provide novel insights into prognostic evaluations and therapeutic strategies for CRC.

## 4. Materials and Methods

### 4.1. The Study Population and Data Collection

Patients with CRC enrolled in two clinical studies investigating chemotherapy-induced peripheral neuropathy at NCKUH were used for the analysis. The information of these two studies was described in detail in the previous study [75]. All patients in these two studies were at stage III or high-risk stage II and underwent standard surgery followed by adjuvant chemotherapy with mFOLFOX6. After the treatment, patients were regularly followed up with computed tomography (CT) scanning to detect recurrence according to the routine daily practice of NCKUH. Clinical information, including the age, sex, primary tumor location, tumor histology, tumor grade, status of the *RAS* mutation, *BRAF* mutation, mismatch repair/microsatellite instability (MMR/MSI), recurrence or not, and survival, were obtained from the medical records. The primary tumor samples of CRC were used for immunohistochemical (IHC) staining and immune-related gene expression analyses. Informed consent was provided by each patient before they were included in these studies and the studies were conducted per the principles of the Declaration of Helsinki. The protocols were approved by the Institutional Review Board of NCKUH (A-ER-103-395 and A-ER-104-0153).

### 4.2. Immunohistochemical Staining of PD-L1 and Immunofluorescent Double Staining

The formalin-fixed and paraffin-embedded (FFPE) primary tumor samples of CRC were used for IHC staining. The monoclonal mouse anti-human PD-L1 antibody (Clone 22C3, Dako, 1:50) was used as the primary antibody, and the procedures were performed with the Bond-Max Automated IHC Stainer (Leica Biosystems Newcastle Ltd., Victoria, Australia) per the following protocol. Four-micrometer sections were cut from the paraffin blocks followed by deparaffinized with xylene and pre-treated with the Epitope Retrieval Solution 2 (EDTA buffer, pH 9.0) at 100 °C for 40 min. Subsequently, the sections were incubated with the primary antibody at room temperature for 90 min. After the staining with the primary antibody, sections were incubated with the polymer at room temperature for 8 min using the Bond Polymer Refine Detection Kit (Leica Biosystems, Newcastle Ltd., Newcastle Upon Tyne, UK) and then developed with 3,3′-diaminobenzidine chromogens for 10 min. Counterstaining was carried out with hematoxylin. For immunofluorescent double staining, Rat anti-CD68 antibody (Novus, Centennial, CO, USA, NBP2-33337) and mouse anti-PD-L1 antibody (Invitrogen, Waltham, MA, USA, #14-5983-82) were used to detect PD-L1 expression on macrophages. Alexa Fluor 594-conjugated secondary antibody (Jackson ImmunoResearch, West Grove, PA, USA, #712-585-150) and Alexa Fluor 488-conjugated secondary antibody (Jackson ImmunoResearch, West Grove, PA, USA, #715-545-150) was used for the staining of CD68 and PD-L1, respectively. To determine PD-L1 expression on T lymphocytes, mouse anti-CD3 antibody (Invitrogen, Waltham, MA, USA, #14-0037-82) and rabbit anti-PD-L1 (GeneTex, Irvine, CA, USA, GTX104763) were applied. Alexa Fluor 594-conjugated secondary antibody (Invitrogen, Waltham, MA, USA, A21207) and Alexa Fluor 488-conjugated secondary antibody (Jackson ImmunoResearch, West Grove, PA, USA, #715-545-150) was used for the staining of CD3 and PD-L1, respectively. Tissue sections were co-stained with 4′-6′-diamidino-2-phenylindole (DAPI) to detect nucleus. The stained slides were excited by laser at 405, 488, and 594 nm, respectively, and immunofluorescent images were captured by the confocal microscope (FV-3000, Olympus, Japan).

### 4.3. Scoring of PD-L1 Expression

The level of PD-L1 expression was assessed by an experienced pathologist who was well-trained in the scoring of PD-L1 expression. According to the PD-L1 IHC 22C3 pharmDx package insert [76], the TPS and CPS were calculated to determine the level of PD-L1 expression. In brief, the TPS was defined as the number of viable tumor cells showing partial or complete membrane staining of PD-L1 divided by the total number of viable tumor cells and then multiplied by 100%. The CPS was defined as the number of PD-L1-positive cells (tumor cells, macrophages, and lymphocytes) divided by the total number of viable tumor cells and then multiplied by 100 [77,78]. At least 100 viable tumor cells are required to evaluate the PD-L1 expression in each testing.

### 4.4. Analysis of Immune-Related Gene Expression

The oncomine immune response research assay was performed on the primary CRC tumor samples to evaluate the immune-related gene expression as previously described [75]. Briefly, the RecoverAll Total Nucleic Acid Isolation Kit (Thermo Fisher Scientific, Waltham, MA, USA) was used to extract the RNA from FFPE primary CRC tumor tissues followed by reverse transcription with 20 ng RNA using the SuperScript IV VILO Master Mix kit. After cDNA synthesis, the Ion AmpliSeq Kit^TM^ for Chef DL8 (Thermo Fisher Scientific, Waltham, MA, USA) was used for the preparation of libraries. Template preparation, chip loading, and sequencing were carried out on the Ion Chef^TM^ System and the Ion S5 XL sequencing system per the manufacturer’s instructions. Torrent Suite (Thermo Fisher Scientific, Waltham, MA, USA) was used to process the raw gene expression data and the DESeq2 package in R was used for the analysis. After data normalization by the DESeq function, the expression levels of genes were compared between the high and low PD-L1 expression groups to identify the differentially expressed genes.

### 4.5. Gene Set Enrichment Analyses

RNA-Seq data of CRC from the TCGA PanCancer Atlas database were downloaded from cBioportal (https://www.cbioportal.org/ (accessed on 11 May 2022)). The data of stage II and III CRCs were extracted to explore the potential pathways related to *CD274* expression. The median expression value of *CD274* was used to divide CRCs into two groups: the high and low *CD274* expression groups. GSEA with the HALLMARK gene set analysis (50 gene sets) was performed using the GSEA software (GSEA v4.23 for windows). The gene sets with nominal *p*-value of <0.05 and a false discovery rate (FDR) q-value of <0.25 were considered the significant enrichment gene sets.

### 4.6. Analysis of Immune Cell Infiltration

Stage II and III CRCs from the TCGA PanCancer Atlas database were divided into the high and low *CXCL9* expression groups per the median *CXCL9* expression levels. RNA expression data were used for the analysis of immune cell infiltration by using the Tumor Immune Estimation Resource 2.0 (TIMER2.0) web server (http://timer.cistrome.org/ (accessed on 20 August 2022)). The immune composition, including the naïve B cells, plasma B cells, CD8+ T cells, CD4+ T cells, follicular helper T cells, regulatory T cells, neutrophils, M1 macrophages, and M2 macrophages, was estimated by the CIBERSORT cellular composition estimation algorithm. The EPIC algorithm was used for characterizing cancer-associated fibroblasts. The infiltration levels of immune cells and cancer-associated fibroblasts were compared between CRCs with high and low *CXCL9* expression.

### 4.7. Statistical Analysis

Statistical analyses were conducted using SPSS version 23.0 (SPSS Inc., Chicago, IL, USA). Categorical data were analyzed using the Chi-squared test or Fischer’s exact test. Spearman’s correlation coefficient was used to determine the correlation of expression levels between different genes. Recurrence-free survival (RFS) was defined as the time from curative surgery to the time of recurrence detected by CT scanning, and OS was defined as the time from curative surgery to death. RFS and OS were illustrated by Kaplan–Meier curves and the log-rank test was used to determine the differences between groups. Kaplan–Meier survival plots were generated by using GraphPad Prism 9. The Cox proportional hazards regression model was performed for univariate and multivariate survival analyses. A *p*-value of <0.05 was considered statistically significant. When comparing the expression of immune-related genes between groups, an adjusted *p*-value of <0.1 was considered statistically significant.

## Figures and Tables

**Figure 1 ijms-23-13277-f001:**
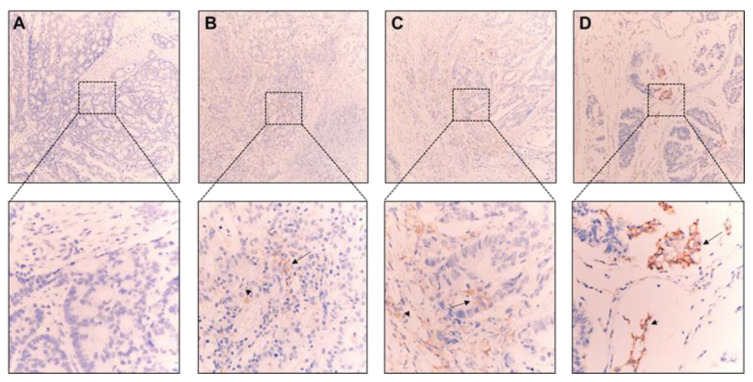
Representative images of PD-L1 staining in CRC tumor cells (arrow head) and immune cells (arrow). CRCs with CPS of <1 (**A**), 1–4 (**B**), 5–9 (**C**), and ≥10 (**D**) are shown at 100× and 400× magnification.

**Figure 2 ijms-23-13277-f002:**
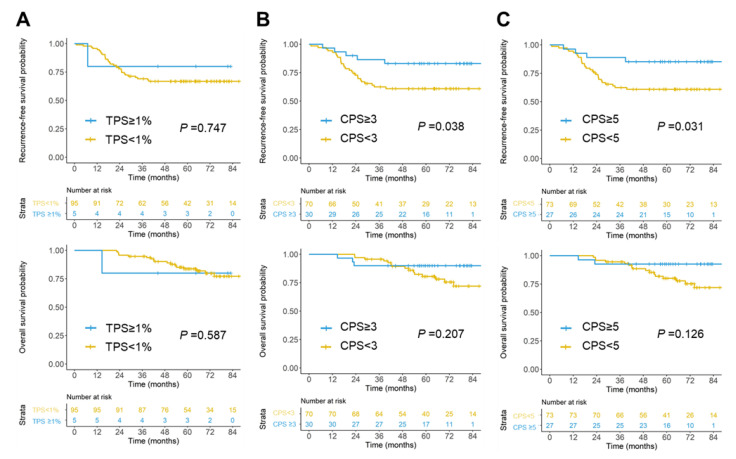
Survival difference of CRC patients at different cutoffs of PD-L1 expression levels. Kaplan–Meier analyses of recurrence-free survival and overall survival in CRC patients with high and low PD-L1 expression using a TPS cutoff value of 1 (**A**), CPS cutoff values of 3 (**B**), and 5 (**C**) are shown. RFS, recurrence-free survival; OS, overall survival; CI, 95% confidence interval.

**Figure 3 ijms-23-13277-f003:**
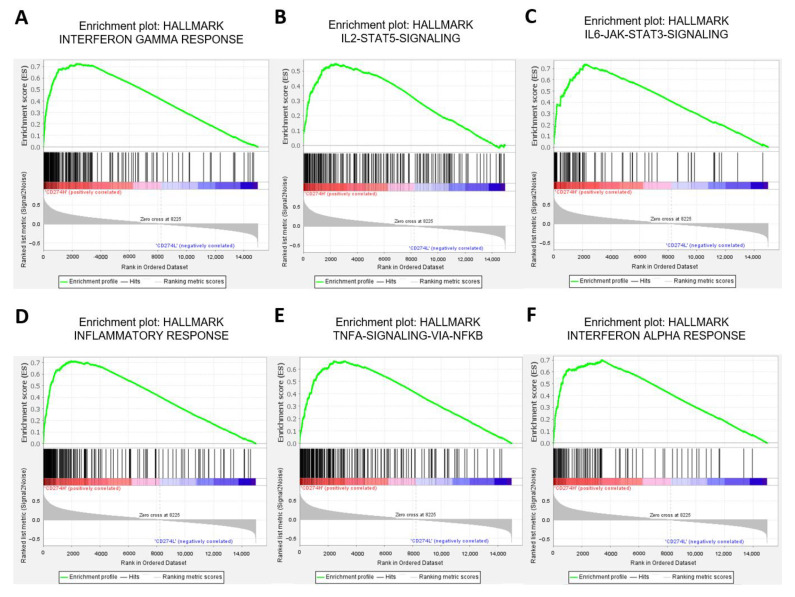
Gene set enrichment analysis in CRCs with high and low *CD274* expression. Hallmark gene sets were applied for gene set enrichment analysis in two phenotypes: CRCs with high and low *CD274* expression according to the medial expression level of *CD274*. The colored band at the bottom represents the degree of correlation of the expression of these genes (red for a positive correlation and blue for a negative correlation) in each gene signature. The immune-related gene sets enriched in CRCs with high *CD274* expression are shown, including (**A**) the IFN-gamma response, (**B**) IL-2-STAT5-signaling, (**C**) IL-6-JAK-STAT-signaling, (**D**) the inflammatory response, (**E**) TNFA-signaling-vial NFKB, and (**F**) the IFN-alpha response. IFN, interferon; TNFA, tumor necrosis factor-alpha; NFKB, nuclear factor kappa-B.

**Figure 4 ijms-23-13277-f004:**
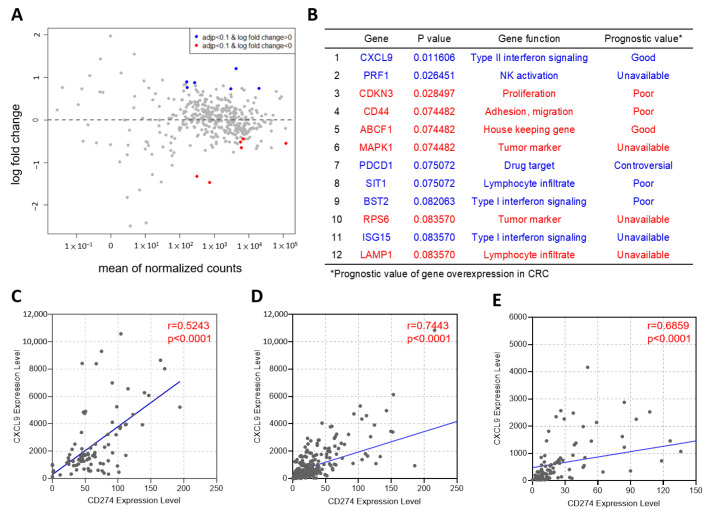
Immune response genes differentially expressed between CRCs with high and low PD-L1 expressions. (**A**) The MA plot shows the log-fold change against the average expression levels of 395 immune-related genes between CRCs with high (CPS ≥ 5) and low PD-L1 expression (CPS < 5). The y-axis represents the log2 fold of change for each gene and the x-axis represents the mean value of normalized reads. Points in red and blue indicate genes with significantly higher and lower expression levels in CRCs with high PD-L1 expression (adjusted *p* < 0.1). (**B**) The list of differentially expressed genes with statistical significance (*p* < 0.1) between the groups of high and low PD-L1 expressions, and the prognostic value of those genes in CRC [7,18,19,20,21,22,23,24,25,26,27,28,29,30,31,32]. Genes in red and blue indicate genes with significantly higher and lower expression levels in CRCs with high PD-L1 expression (adjusted *p* < 0.1). The correlation between *CD274* and *CXCL9* was analyzed using the Spearman correlation coefficient in CRCs from NCKUH (**C**), and the colon (**D**) and rectal cancer (**E**) cohort from the TCGA PanCancer Atlas. r, Spearman’s rank correlation coefficient.

**Figure 5 ijms-23-13277-f005:**
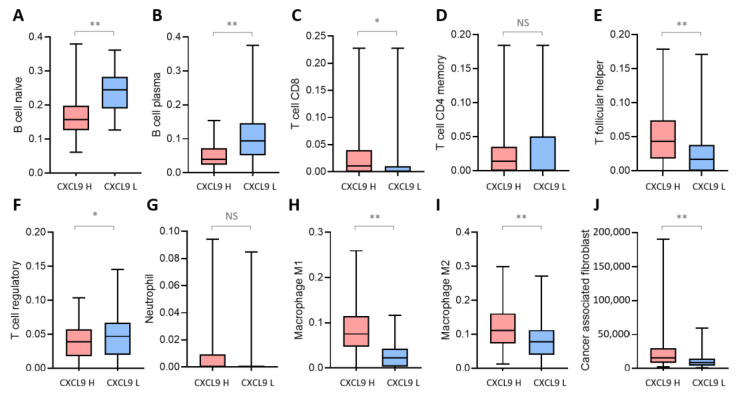
Immune cell infiltrations in CRCs with high and low *CXCL9* expression. Immune cells infiltrations, including naïve B cells (**A**), plasma cells (**B**), CD8+ T cells (**C**), CD4+ memory T cells (**D**), T follicular helper cells (**E**), regulatory T cells (**F**), neutrophils (**G**), M1 macrophages (**H**), M2 macrophages (**I**), and cancer associated fibroblasts (**J**), were estimated using TIMER 2.0 and the *t*-test was used to determine the difference between CRCs with high and low *CXCL9* expression. The central line, bottom, and top of the box indicate the median, 25th percentile, and 75th percentiles of the data. ** *p* < 0.001; * *p* < 0.05; NS, not significant.

**Table 1 ijms-23-13277-t001:** Clinical characteristics of 100 CRC patients.

Characteristics	No. of Cases (%)
Total patients	100 (100)
Age, median (range)—year	56.5 (29–78)
Gender	
Male	54 (54)
Female	46 (46)
Primary tumor location	
Right-sided	23 (23)
Left-sided ^a^	76 (76)
Multiple	1 (1)
Tumor histology	
Adenocarcinoma	91 (91)
Mucinous adenocarcinoma	9 (9)
Tumor grade	
Well differentiated	5 (5)
Moderately differentiated	91 (91)
Poorly differentiated	4 (4)
TMN stage	
II	3 (3)
III	97 (97)
*RAS* status	
Mutant	42 (42)
*KRAS*	40 (40)
*NRAS*	2 (2)
Wild type	58 (58)
*BRAF* status	
Mutant	6 (6)
Wild type	94 (94)
MMR status	
dMMR/MSI-high	6 (6)
pMMR/MSI-stable and -low	94 (94)
PD-L1 expression	
Tumor proportion score	
<1%	95 (95)
≥1%	5 (5)
Combined positive score	
<1	47 (47)
1–4	26 (26)
5–9	15 (15)
≥10	12 (12)
Recurrence	
Yes	32 (32)
No	68 (68)

^a^ The left-sided colon is defined as the splenic flexure to rectum. Abbreviations: MMR, mismatch repair; dMMR, deficient mismatch repair; MSI, microsatellite instability.

**Table 2 ijms-23-13277-t002:** Clinical characteristics of patients with high and low PD-L1 expression.

Characteristics	PD-L1 Expression		*p*-Value
	CPS ≥ 5	CPS < 5	
Age—no. (%)			
≥65 years	7 (7)	20 (20)	0.883
<65 years	20 (20)	53 (53)	
Gender—no. (%)			
Male	16 (16)	38 (38)	0.521
Female	11 (11)	35 (35)	
Tumor histology—no. (%)			
Adenocarcinoma	25 (25)	66 (66)	0.735
Mucinous adenocarcinoma	2 (2)	7 (7)	
Tumor grade—no. (%)			
Well differentiated	0 (0)	5 (5)	0.231
Moderately differentiated	25 (25)	66 (66)	
Poorly differentiated	2 (2)	2 (2)	
Tumor location			
Primary tumor location—no. (%)			
Right-sided	10 (10)	13 (13) ^#^	0.046 *
Left-sided	17 (17)	59 (60) ^#^	
Stage—no. (%)			
II	2 (2)	1 (1)	0.116
III	25 (25)	72 (72)	
*RAS* status—no. (%)			
Mutant	12 (12)	30 (30)	0.763
Wild type	15 (15)	43 (43)	
*BRAF* status—no. (%)			
Mutant	1 (1)	5 (5)	0.557
Wild type	26 (26)	68 (68)	
MMR status—no. (%)			
dMMR/MSI-high	4 (4)	2 (2)	0.024 *
pMMR/MSI-stable or -low	23 (23)	71 (71)	

Abbreviation: MMR, mismatch repair; dMMR, deficient mismatch repair; MSI, microsatellite instability. * *p* < 0.05. ^#^ One patient was diagnosed of simultaneous right- and left-sided CRC, and this case was not included in the comparison study.

**Table 3 ijms-23-13277-t003:** Univariate and multivariate analysis for RFS.

Characteristics	Univariate Analysis			Multivariate Analysis		
	Hazard Ratio	95% CI	*p*-Value	Hazard Ratio	95% CI	*p*-Value
Age						
≥65 years	1.703	0.832–3.487	0.145	1.540	0.751–3.157	0.238
<65 years	-			-	-	
Gender						
Male	0.818	0.409–1.636	0.57			
Female						
Tumor histology						
Adenocarcinoma	0.592	0.207–1.690	0.327			
Mucinous adenocarcinoma						
Tumor grade						
Well differentiated			0.787			
Moderately differentiated	1.544	0.210–11.341	0.669			
Poorly differentiated	2.266	0.205–25.008	0.504			
Tumor location						
Right-sided	0.649	0.249–1.690	0.376			
Left-sided						
Stage						
II	0.047	0.000–173.959	0.466			
III						
*RAS* status						
Mutant	2.338	1.154–4.737	0.018 *	2.307	1.136–4.682	0.021 *
Wild type						
*BRAF* status						
Mutant	0.044	0.000–15.846	0.299			
Wild type						
MMR status						
dMMR/MSI-high	0.484	0.066–3.544	0.475			
pMMR/MSI-stable or -low						
PD-L1 expression						
CPS ≥ 5	0.330	0.116–0.942	0.038 *	0.327	0.115–0.934	0.037 *
CPS < 5						

Abbreviation: CPS, combined positive score; dMMR, deficient mismatch repair; pMMR, proficient mismatch repair; MMR, mismatch repair; MSI, microsatellite instability. * *p* < 0.05.

## Data Availability

Publicly available datasets were analyzed in this study. These data can be found on this website: https://www.cbioportal.org/study/summary?id=coaread_tcga_pan_can_atlas_2018 (accessed on 14 September 2022). Our own data presented in this study are available from the corresponding author upon reasonable request.

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
