# Peer review of "PD-L1 Expression in High-Risk Early-Stage Colorectal Cancer—Its Clinical and Biological Significance in Immune Microenvironment"

_ijms, 2022, doi:10.3390/ijms232113277_

Round 1

Reviewer 1 Report

The authors present a study correlating PDL1 expression with free recurrence survival in a Taiwanese cohort of CRC patients.

In my opinion some points need to be revised:

Figure1 and methods:

The authors are encouraged to provide the images at lower magnification in order to appreciate the organization of the tumor thus showing the marginal region in which there is the accumulation of immune cells. The authors describe in the methods that CPS is calculated considering the expression of PDL1 in tumor cells as in immune cells. It could be clearer if the authors indicate in the staining (I.E. WITH ARROWS) a membranous positivity for tumor cells and a cytoplasmic one for immune cells: in this case the higher magnification may be informative.

The panel C and D look too similar, so the CPS score is not indicative.

A study from Kim HR et al Scientific Reports 2016 reported that the PDL1 correlates with good prognosis only if it’s expressed on immune cells. Even if this paper is about a different type of cancer, I think that the authors can address this point with a double staining identifying immune cells (i.e. CD3 antibody for lymphocytes and CD68 for macrophages).

Generally, I suggest to better describe the PDL1 staining and analysis in the methods section also citing the PD-L1 IHC 22C3 interpretation guide from Agilent.

Results section:

Paragraph 2.4

The authors indicate the immune related pathways enriched in the high PDL1 expressing group. They are encouraged to deeper discuss each of these in terms of studies reported in CRC and, in case, their correlation with clinical outcome.

Paragraph 2.5 and Figure 4

The authors presented the differential gene expressions were compared between patients with high (CPS ≥ 5) and low PD-L1 expression (CPS < 5) and they listed the more indicative genes in a table (panel B -Figure4). I suggest adding in the table, if already known, a role for each gene in CRC (correlation with survival, prognostic factor…) and reporting the relative reference.

Paragraph 2.6

In lines 204-206 the authors claim the increase infiltration of CD8 T cells at the expense of Treg but also an enrichment in M1, M2 and CAF in the group of high CXCL9 expression. Given the different roles of these immune populations in tumor microenvironment, I think that a discussion is needed to explain this result. For instance CAFs has been reported to block CD8 infiltration and to confer resistance to immune checkpoint blockade (DOI: 10.1158/0008-5472.CAN-21-4141): this aspect is needed to argue.

Discussion:

The authors reported that CXCL9 is the gene with higher differential expression between the PDL1 categories and that correlates with higher immune infiltration. Even the authors in line 325 mentioned a missing point in the paper they are encouraged to at least discuss the possible underlying mechanism (i.e. the CXCR3.CXCL9 axis reviewed in https://doi.org/10.1016/j.immuni.2019.05.013).

Concluding I think that the paper can benefit of some new experimental data:  in addition to that already suggested the identification of CXCL9 in the samples should be done. Moreover, major studies in terms of tumor microenvironment are needed otherwise the conclusions are not fully demonstrated.

Reviewer 2 Report

Dear Authors, 

please, find my comments on the manuscript below:

A.d. 2.2. Survival Analysis by Different Levels of PD-L1 Expression

Figure 2.:The number of cases for the compared subgroups in the analysis with the CPS cut-off value> = 3 / <3 should be reported. It cannot, as for the cut-off value CPS> = 5 / <5, be deduced from Table 1.

A.d. 2.3. Clinicopathological Features of Patients with high and low PD-L1 Expression

Table 2: It would be advantageous to provide the percentages calculated for columns or rows (within a given parameter) in addition to the absolute counts. This would make data easier to interpret.

In the case of the "primary tumour location" parameter, the total number of patients in CPS <5 to 72. It should be 73.

Table 3. Redundant dot in the HR value for Age > = 65 yrs - 1..540

A.d. 2.4. Immune Pathways Associated with High PD-L1 Expression 

It was stated that 16 gene signatures were statistically enriched in the high CD274 expression group (lines 148-9). In the Materials and Methods section, subsection 4.5. Gene Set Enrichment Analysis, the conditions of significant enrichment were given as a nominal p-value of <0.05 AND a false discovery rate (FDR) q-value of <0.25. In Supplementary Table 2, APICAL_JUNCTION gene set had a p-value of 0.123, so, it did not meet the assumed conditions. This inaccuracy should be cleared up. 

Round 2

Reviewer 1 Report

The authors addressed all requested points. 

Reviewer 2 Report

Dear Authors, 

I've read your replys to revievers and I have no further comments.